# GIT: A Generative Image-to-text Transformer for Vision and Language

**Jianfeng Wang** *jianfw@microsoft.com*
**Zhengyuan Yang** *zhengyang@microsoft.com*
**Xiaowei Hu** *xiaowei.hu@microsoft.com*
**Linjie Li** *lindsey.li@microsoft.com*
**Kevin Lin** *keli@microsoft.com*
**Zhe Gan** *zhe.gan@microsoft.com*
**Zicheng Liu** *zliu@microsoft.com*
**Ce Liu** *ce.liu@microsoft.com*
**Lijuan Wang** *lijuanw@microsoft.com*
*Microsoft Cloud and AI*

**Reviewed on OpenReview:** *https: // openreview. net/ forum? id= b4tMhpNOJC*

## Abstract

In this paper, we design and train a **G**enerative **I**mage-to-text **T**ransformer, GIT, to unify vision-language tasks such as image/video captioning and question answering. While generative models provide a consistent network architecture between pre-training and fine-tuning, existing work typically contains complex structures (uni/multi-modal encoder/decoder) and depends on external modules such as object detectors/taggers and optical character recognition (OCR). In GIT, we simplify the architecture as one image encoder and one text decoder under a single language modeling task. We also scale up the pre-training data and the model size to boost the model performance. Without bells and whistles, our GIT establishes new state of the arts on numerous challenging benchmarks with a large margin. For instance, our model surpasses the human performance for the first time on TextCaps (138.2 vs. 125.5 in CIDEr). Furthermore, we present a new scheme of generation-based image classification and scene text recognition, achieving decent performance on standard benchmarks.

## 1 Introduction

Table 1: Comparison with prior SOTA on image/video captioning and question answering (QA) tasks. *: evaluated on the public server. CIDEr scores are reported for Captioning tasks. Prior SOTA: COCO(Zhang et al., 2021a), nocaps (Yu et al., 2022), VizWiz-Caption (Gong et al., 2021), TextCaps (Yang et al., 2021c),ST-VQA (Biten et al., 2022),VizWiz-VQA (Alayrac et al., 2022),OCR-VQA (Biten et al., 2022),MSVD (Lin et al., 2021),MSRVTT (Seo et al., 2022),VATEX (Tang et al., 2021),TVC (Tang et al., 2021),MSVD-QA (Wang et al., 2022a),TGIF-Frame (Zellers et al., 2021),Text Recog. (Lyu et al., 2022). Details of GIT2 are presented in supplementary materials.

| | Image captioning | | | | Image QA | | | Video captioning | | | | Video QA | | Text Rec. |
| --- | --- | --- | --- | --- | --- | --- | --- | --- | --- | --- | --- | --- | --- | --- |
| | COCO* | nocaps* | VizWiz* | TextCaps* | ST-VQA* | VizWiz* | OCR-VQA | MSVD | MSRVTT | VATEX* | TVC* | MSVD-QA | TGIF-Frame | Avg on 6 |
| Prior SOTA[1] | 138.7 | 120.6 | 94.1 | 109.7 | 69.6 | 65.4 | 67.9 | 120.6 | 60 | 86.5 | 64.5 | 48.3 | 69.5 | 93.8 |
| GIT (ours) | 148.8 | 123.4 | 114.4 | 138.2 | 69.6 | 67.5 | 68.1 | 180.2 | 73.9 | 93.8 | 61.2 | 56.8 | 72.8 | 92.9 |
| $\Delta$ | +10.1 | +2.8 | +20.3 | +28.5 | +0.0 | +2.1 | +0.2 | +59.6 | +13.9 | +7.3 | -3.3 | +8.5 | +3.3 | -0.9 |
| GIT2 (ours) | 149.8 | 124.8 | 120.8 | 145.0 | 75.8 | 70.1 | 70.3 | 185.4 | 75.9 | 96.6 | 65.0 | 58.2 | 74.9 | 94.5 |
| $\Delta$ | +11.1 | + 4.2 | +26.7 | +35.3 | +6.2 | +4.7 | +2.4 | +64.8 | +15.9 | +10.1 | +0.5 | +9.9 | +5.4 | +0.7 |

---

[1]Prior SOTA: among all the numbers reported in publications before 8/2022, as far as we know.

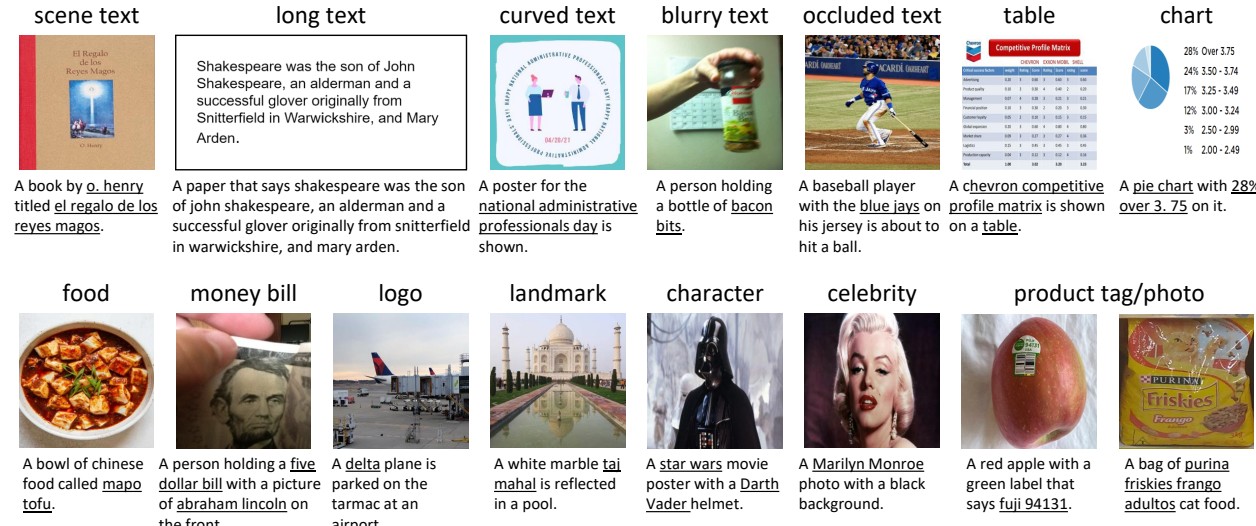

Figure 1: Example captions generated by GIT. The model demonstrates strong capability of recognizing scene text, tables/charts, food, banknote, logos, landmarks, characters, products, *etc.*

Tremendous advances have been made in recent years on vision-language (VL) pre-training, especially based on the large-scale data of image-text pairs, *e.g.*, CLIP (Radford et al., 2021), Florence (Yuan et al., 2021), and SimVLM (Wang et al., 2021b). The learned representation greatly boosts the performance on various downstream tasks, such as image captioning (Lin et al., 2014), visual question answering (VQA) (Goyal et al., 2017), and image-text retrieval.

During pre-training, Masked Language Modeling (MLM) and Image-Text Matching (ITM) tasks have been widely used (Wang et al., 2020; Fang et al., 2021c; Li et al., 2020; Zhang et al., 2021a; Chen et al., 2020b; Dou et al., 2021; Wang et al., 2021a; Kim et al., 2021). However, these losses are different from the downstream tasks, and task-specific adaptation has to be made. For example, ITM is removed for image captioning (Wang et al., 2021a; Li et al., 2020), and an extra randomly initialized multi-layer perceptron is added for VQA (Wang et al., 2021b; Li et al., 2020). To reduce this discrepancy, recent approaches (Cho et al., 2021; Wang et al., 2021b; Yang et al., 2021b; Wang et al., 2022b) have attempted to design unified generative models for pre-training, as most VL tasks can be cast as generation problems. These approaches typically leverage a multi-modal encoder and a text decoder with careful design on the text input and the text target. To further push the frontier of this direction, we present a simple Generative Image-to-text Transformer, named GIT, which consists only of one image encoder and one text decoder. The pre-training task is just to map the input image to the entire associated text description with the language modeling objective. Despite its simplicity, GIT achieves new state of the arts across numerous challenging benchmarks with a large margin, as summarized in Table 1.

The image encoder is a Swin-like vision transformer (Dosovitskiy et al., 2021; Yuan et al., 2021) pre-trained on massive image-text pairs based on the contrastive task (Jia et al., 2021; Radford et al., 2021; Yuan et al., 2021). This eliminates the dependency on the object detector, which is used in many existing approaches (Anderson et al., 2018; Li et al., 2020; Wang et al., 2020; Zhang et al., 2021a; Chen et al., 2020b; Fang et al., 2021c). To extend it to the video domain, we simply extract the features of multiple sampled frames and concatenate them as the video representation. The text decoder is a transformer network to predict the associated text. The entire network is trained with the language modeling task. For VQA, the input question is treated as a text prefix, and the answer is generated in an auto-regressive way. Furthermore, we present a new generation-based scheme for ImageNet classification, where the predicted labels come directly from our generative model without pre-defining the vocabulary.

The approach is simple, but the performance is surprisingly impressive after we scale up the pre-training data and the model size. Fig. 1 shows captions generated by the GIT fine-tuned with TextCaps. The samples

demonstrate the model's strong capability of recognizing and describing scene text, tables, charts, food, banknote, logos, landmarks, characters, celebrities, products, *etc.*, indicating that our GIT model has encoded rich multi-modal knowledge about the visual world.

Our main contributions are as follows.

- We present GIT, which consists of only one image encoder and one text decoder, pre-trained on 0.8 billion image-text pairs with the language modeling task.

- We demonstrate new state-of-the-art performance over numerous tasks on image/video captioning and QA (Table 1), without the dependency on object detectors, object tags, and OCR. On TextCaps, we surpass the human performance for the first time. This implies that a simple network architecture can also achieve strong performance with scaling.

- We demonstrate that GIT pre-trained on the image-text pairs is capable of achieving new state-of-the-art performance even on video tasks without video-dedicated encoders.

- We present a new scheme of generation-based image classification. On ImageNet-1K, we show a decent performance (88.79% top-1 accuracy) with our GIT.

## 2 Related Work

In VL pre-training, multi-task pre-training has been widely used to empower the network with multiple or enhanced capabilities. For example, MLM and ITM are widely adopted pre-training tasks (Li et al., 2020; Kim et al., 2021; Zhang et al., 2021a; Wang et al., 2020; Xue et al., 2021; Lu et al., 2019; Tan & Bansal, 2019). Recently, the image-text contrastive loss has also been added in Yu et al. (2022); Li et al. (2021a); Wang et al. (2021a). Since most VL tasks can be formulated as the text generation task (Cho et al., 2021), a single generation model can be pre-trained to support various downstream tasks. The input and output texts are usually carefully designed to pre-train such a generation model. For example in Cho et al. (2021), the text is properly masked as the network input and the goal is to recover the masked text span. SimVLM (Wang et al., 2021b) randomly splits a text sentence into the input and the target output. In these methods, a multi-modal transformer encoder is utilized to incorporate the text inputs before decoding the output.

For image representation, Faster RCNN has been used in most existing approaches (Anderson et al., 2018; Li et al., 2020; Wang et al., 2020; Zhang et al., 2021a; Chen et al., 2020b; Fang et al., 2021c) to extract the region features. Recently, a growing interest is in dense representation (Huang et al., 2020; Wang et al., 2021b;a; Kim et al., 2021; Fang et al., 2021b; Dou et al., 2021; Li et al., 2021a) from the feature map, which requires no bounding box annotations. Meanwhile, it is easy to train the entire network in an end-to-end way. In addition to the representation from the feature map, object tags (Li et al., 2020; Wang et al., 2020; Zhang et al., 2021a; Cornia et al., 2021; Fang et al., 2021b) are leveraged to facilitate the transformer to understand the context, especially the novel objects. For scene-text-related tasks, OCR is invoked to generate the scene text as additional network input, *e.g.*, in Hu et al. (2020); Yang et al. (2021c). For the text prediction, A transformer network is typically used, which can incorporate the cross-attention module to fuse the image tokens, *e.g.*, Cho et al. (2021); Alayrac et al. (2022); Yang et al. (2021b); Yu et al. (2022), or only the self-attention modules where the image tokens are concatenated with the text tokens, *e.g.*, Li et al. (2020); Chen et al. (2020b); Zhang et al. (2021a); Wang et al. (2020); Fang et al. (2021b).

Along the direction of scaling on VL tasks, LEMON (Hu et al., 2021a) studies the behavior of the detector-based captioning model with MLM. CoCa (Yu et al., 2022) studies different model sizes, but on the same pre-training data. In this paper, we present a comprehensive study on 9 various benchmarks (3 in main paper and 6 in supplementary materials, image/video captioning & QA tasks) with 3 different model sizes and 3 different pre-training data scales (9 data points for each benchmark).

## 3 Generative Image-to-text Transformer

With large-scale image-text pairs, our goal is to pre-train a VL model which is simple yet effective to benefit image/video captioning and QA tasks. As the input is the image and the output is the text, the minimal set

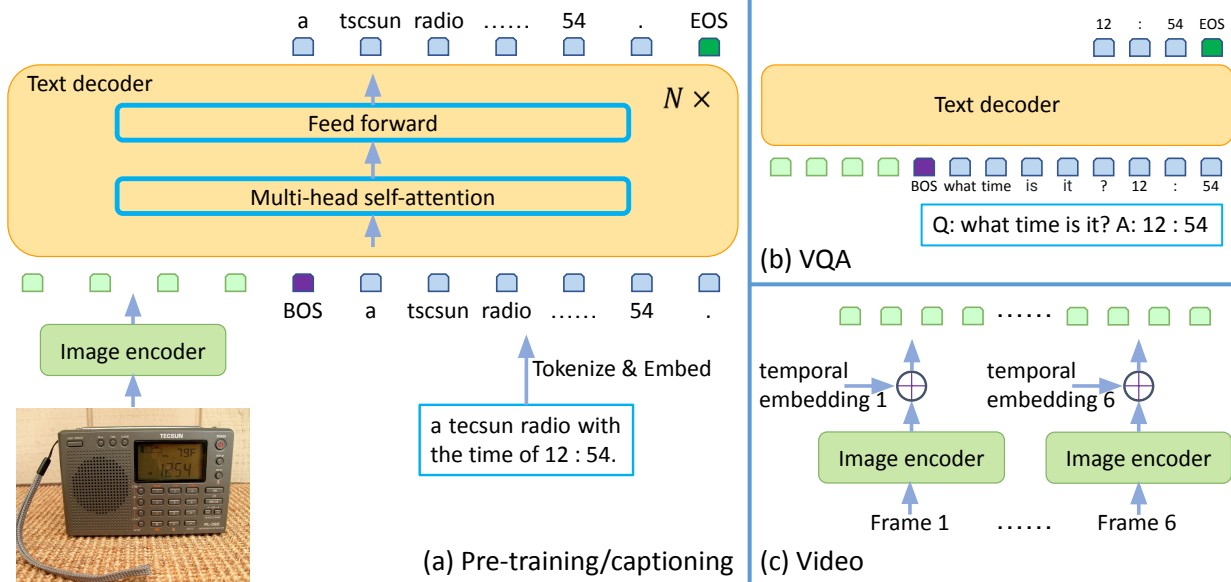

Figure 2: Network architecture of our GIT, composed of one image encoder and one text decoder. (a): The training task in both pre-training and captioning is the language modeling task to predict the associated description. (b): In VQA, the question is placed as the text prefix. (c): For video, multiple frames are sampled and encoded independently. The features are added with an extra learnable temporal embedding (initialized as 0) before concatenation.

of components could be one image encoder and one text decoder, which are the only components of our GIT as illustrated in Fig. 2.

## 3.1 Network Architecture

The image encoder is based on the contrastive pre-trained model (Yuan et al., 2021). The input is the raw image and the output is a compact 2D feature map, which is flattened into a list of features. With an extra linear layer and a layernorm layer, the image features are projected into $D$ dimensions, which are the input to the text decoder. We use the image encoder pre-trained with contrastive tasks because recent studies show superior performance with such image encoder, e.g. Yuan et al. (2021); Dou et al. (2021); Alayrac et al. (2022). In Sec 4.6 and supplementary materials, we also observe the VL performance boosts significantly with a stronger image encoder. This is consistent with the observation in object detection-based approaches, e.g. in Wang et al. (2020); Zhang et al. (2021a). The concurrent work of CoCa (Yu et al., 2022) unifies the contrastive task and the generation task. as one pre-training phase. Our approach is equivalent to separating the two tasks sequentially: ($i$) using the contrastive task to pre-train the image encoder followed by ($ii$) using the generation task to pre-train both the image encoder and text decoder.

The text decoder is a transformer module to predict the text description. The transformer module consists of multiple transformer blocks, each of which is composed of one self-attention layer and one feed-forward layer. The text is tokenized and embedded into $D$ dimensions, followed by an addition of the positional encoding and a layernorm layer. The image features are concatenated with the text embeddings as the input to the transformer module. The text begins with the [BOS] token, and is decoded in an auto-regressive way until the [EOS] token or reaching the maximum steps. The seq2seq attention mask as in Fig. 3 is applied such that the text token only depends on the preceding tokens and all image tokens, and image tokens can attend to each other. This is different from a unidirectional attention mask, where not every image token can rely on all other image tokens.

Instead of well initializing the image encoder, we randomly initialize the text decoder. This design choice is highly motivated from the experiment studies of Wang et al. (2020), in which the random initialization shows

similar performance, compared with the BERT initialization. This could be because the BERT initialization cannot understand the image signal, which is critical for VL tasks. Without dependency of the initialization, we can easily explore different design choices. The concurrent work of Flamingo (Alayrac et al., 2022) employs a similar architecture of image encoder + text decoder, but their decoder is pre-trained and frozen to preserve the generalization capability of the large language model. In our GIT, all parameters are updated to better fit the VL tasks.

An alternative architecture is the cross-attention-based decoder to incorporate the image signals instead of concatenation with self-attention. Empirically as shown in supplementary material (Appendix G.2), with large-scale pre-training, we find the self-attention-based decoder achieves better performance overall, while in small-scale setting, the cross-attention-based approach wins. A plausible explanation is that with sufficient training, the decoder parameters can well process both the image and the text, and the image tokens can be better updated with the self-attention for text generation. With cross-attention, the image tokens cannot attend to each other.

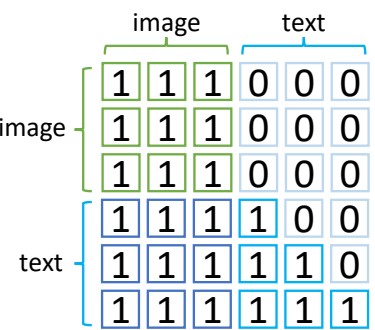

Figure 3: `seq2seq` attention mask is applied to the transformer. If $(i, j)$ is 1, the $i$-th output can depend on the $j$-th input; otherwise, not.

### 3.2 Pre-training

For each image-text pair, let $I$ be the image, $y_i, i \in \{1, \cdots, N\}$ be the text tokens, $y_0$ be the `[BOS]` token and $y_{N+1}$ be the `[EOS]` token. We apply the language modeling (LM) loss to train the model. That is,

$$l = \frac{1}{N+1} \sum_{i=1}^{N+1} \text{CE}(y_i, p(y_i | I, \{y_j, j = 0, \cdots, i-1\})), \tag{1}$$

where CE is the cross-entropy loss with label smoothing of 0.1.

An alternative choice is MLM, which predicts typically 15% of input tokens in each iteration. To predict all tokens, we have to run at least $1/0.15 = 6.7$ epochs. For LM, each iteration can predict all tokens, which is more efficient for large-scale pre-training data. In Hu et al. (2021a), the ablation studies also show that LM can achieve better performance with limited epochs. In our large-scale training, the number of epoch is only 2 due to computational resource limitation, and thus we choose LM. Meanwhile, most of the recent large-scale language models are also based on LM, e.g. Brown et al. (2020); Chowdhery et al. (2022).

Without the image input, the model is reduced to a decoder-only language model, similar to GPT3 (Brown et al., 2020) in the architecture wise. Thus, this design also enables the possibility to leverage the text-only data to enrich the decoding capability with a scaled-up decoder. We leave this as future work.

### 3.3 Fine-tuning

For the image captioning task, as the training data format is the same as that in pre-training, we apply the same LM task to fine-tune our GIT.

For visual question answering, the question and the ground-truth answer are concatenated as a new special caption during the fine-tuning, but the LM loss is only applied on the answer and the `[EOS]` tokens. During inference, the question is interpreted as the caption prefix and the completed part is the prediction. Compared with the existing approaches (Wang et al., 2021a;b; Zhang et al., 2021a; Li et al., 2022b) for VQAv2 (Goyal et al., 2017), our model is generative without pre-defining the candidate answers, even in inference. This imposes more challenges as the model has to predict at least two correct tokens: one for the answer and another for `[EOS]`. In contrast, the existing work pre-collects the answer candidate, recasts the problem as a classification problem, and only needs to predict once. However, considering the benefit of the free-form answer, we choose the generative approach. Due to difficulty of the generative model, we observe slightly worse performance on VQAv2 than the discriminative existing work. For the scene-text related VQA tasks, existing approaches (Yang et al., 2021c; Hu et al., 2020) typically leverages the OCR engine to generate the

scene text and use dynamic pointer network to decide the current output token should be OCR or the general text. Here, our approach depends on no OCR engine, and thus no dynamic pointer network. Empirically, we find the model gradually learns how to read the scene text with large-scale pre-training, and our model achieves new SoTA performance on these tasks.

Our model is not specifically designed for the video domain, but we find our model can also achieve competitive or even new SOTA performance with a simple architecture change. That is, we sample multiple frames from each video clip, and encode each frame via the image encoder independently. Afterwards, we add a learnable temporal embedding (initialized as zeros), and concatenate the features from sampled frames. The final representation is used in a similar way as the image representation for captioning and question answering.

We also apply our generation model to the image classification task, where the class names are interpreted as image captions, and our GIT is fine-tuned to predict the result in an auto-regressive way. This is different from existing work which normally pre-defines the vocabulary and uses a linear layer (with softmax) to predict the likelihood of each category. This new generation-based scheme is beneficial when new data and new categories are added to the existing dataset. In this case, the network can continuously train on the new data without introducing new parameters.

## 4 Experiments

### 4.1 Setting

We collect 0.8B image-text pairs for pre-training, which include COCO (Lin et al., 2014), Conceptual Captions (CC3M) (Sharma et al., 2018), SBU (Ordonez et al., 2011), Visual Genome (VG) (Krishna et al., 2016), Conceptual Captions (CC12M) (Changpinyo et al., 2021), ALT200M (Hu et al., 2021a), and an extra 0.6B data following a similar collection procedure in Hu et al. (2021a). The image encoder is initialized from the pre-trained contrastive model (Yuan et al., 2021). The hidden dimension ($D$) is 768. The text decoder consists of 6 randomly-initialized transformer blocks. The total number of model parameters is 0.7 billion. The learning rates of the image encoder and the decoder are $1e^{-5}$ and $5e^{-5}$, respectively, and follow the cosine decay to 0. The total number of epochs is 2. During inference, the beam size is 4 and the length penalty (Wu et al., 2016) is 0.6 by default.

Supplementary materials show results on two smaller model variants ($\text{GIT}_B$ and $\text{GIT}_L$) and one even larger model (GIT2) with full details. When comparing with existing approaches, the reference numbers are the best one reported in the corresponding paper unless explicitly specified.

### 4.2 Results on Image Captioning and Question Answering

We comprehensively evaluate the captioning performance on the widely-used Karpathy split (Karpathy & Li, 2015) of COCO (Lin et al., 2014) and Flickr30K (Young et al., 2014), the COCO test set, nocaps (Agrawal et al., 2019)[2] which focuses on novel objects, TextCaps (Sidorov et al., 2020) which focuses on scene-text understanding, and VizWiz-Captions (Gurari et al., 2020) which focuses on the real use case by the vision-impaired people. The results in CIDEr (Vedantam et al., 2015) are shown in Table 2 and 3. From the results, we can see our model achieves the new SOTA performance on all these metrics except on COCO Karpathy test. On nocaps, compared with CoCa (Yu et al., 2022), our model is much smaller in the model size (0.7B vs 2.1B), but achieves higher performance (123.0 vs 120.6 in CIDEr). On Textcaps, our solution outperforms the previous SOTA (TAP Yang et al. (2021c)) by a breakthrough margin (28.5 points in CIDEr), and also surpasses the human performance for the first time. For zero/few-shot evaluation as shown in Table 3, our model can significantly benefit from more shots. With 32-shots, our approach is also better than Flamingo.

On VQA, the evaluation benchmarks include VQAv2 (Goyal et al., 2017), TextVQA (Singh et al., 2019), VizWiz-VQA (Gurari et al., 2018). ST-VQA (Biten et al., 2019), and OCR-VQA (Mishra et al., 2019). Before fine-tuning the model, we run an intermediate fine-tuning on the combination of the training data of VQAv2, TextVQA, ST-VQA, OCR-VQA, VizWiz-VQA, Visual Genome QA (Krishna et al., 2016), GQA (Hudson &

---

[2]We compare all approaches including using external image-text datasets.

Table 2: Results on image captioning. *: the nubmers are from Sidorov et al. (2020); CE: cross-entropy optimization. All numbers are CIDEr scores, and other metrics are shown in supplementary materials. #: winner entry of the CVPR 2021 workshop challenge Anc.-Cap.: Xu et al. (2021) AoANet: Huang et al. (2019) BUTD: Anderson et al. (2018), CoCa: Yu et al. (2022), DistillVLM: Fang et al. (2021c), Flamingo: Alayrac et al. (2022), Human: Agrawal et al. (2019), LEMON: Hu et al. (2021a), M4C-Cap.: Hu et al. (2020) MiniVLM: Wang et al. (2020), MTMA: Gong et al. (2021), OFA: Wang et al. (2022b), OSCAR: Li et al. (2020), UFO: Wang et al. (2021a), UniversalCap: (Cornia et al., 2021) ViTCap: Fang et al. (2021b), VinVL: Zhang et al. (2021a), VIVO: Hu et al. (2021b) SimVLM: Wang et al. (2021b), TAP: Yang et al. (2021c).

| Method | CE |
|---|---|
| MiniVLM | 119.8 |
| DistillVLM | 120.8 |
| ViTCap | 125.2 |
| OSCAR | 127.8 |
| VinVL | 130.8 |
| UFO | 131.2 |
| Flamingo | 138.1 |
| LEMON | 139.1 |
| SimVLM | 143.3 |
| CoCa | 143.6 |
| OFA | **145.3** |
| GIT | 144.8 |

(a) COCO Karp.

| Method | C |
|---|---|
| BUTD | 120.5 |
| VinVL | 138.7 |
| GIT | **148.8** |

(b) COCO test (c40)

| Method | test-std |
|---|---|
| MTMA | 94.1 |
| GIT | **114.4** |

(c) VizWiz-Captions

| Method | Test |
|---|---|
| OSCAR | 80.9 |
| Human | 85.3 |
| VIVO | 86.6 |
| VinVL | 92.5 |
| UFO | 92.3 |
| SimVLM | 115.2 |
| LEMON | 114.3 |
| UniversalCap | 119.3 |
| CoCa | 120.6 |
| GIT | **123.4** |

(d) nocaps

| Method | Test |
|---|---|
| BUTD* | 33.8 |
| AoANet* | 34.6 |
| M4C-Cap.* | 81.0 |
| Anc.-Cap. | 87.4 |
| TAP | 103.2 |
| TAP# | 109.7 |
| Human | 125.5 |
| GIT | **138.2** |

(e) TextCaps

Table 3: Zero/Few/Full-shot evaluation on Flickr30K with Karpathy split.

| Shot | 0 | 16 | 32 | 290 (1%) | full |
|---|---|---|---|---|---|
| Zhou et al. (2020) | - | - | - | - | 68.5 |
| Flamingo | 67.2 | 78.9 | 75.4 | - | - |
| GIT | 49.6 | 78.0 | 80.5 | 86.6 | 98.5 |

Manning, 2019), and OK-VQA (Marino et al., 2019). To avoid data contamination, we remove the duplicate images of the test and validation set of the target benchmarks. As illustrated in Table 4, we achieve new SOTA on VizWiz-VQA and OCR-VQA, and same performance with prior SOTA of LaTr (Biten et al., 2022) on ST-VQA. Compared with the concurrent work of Flamingo (Alayrac et al., 2022), we achieve higher accuracy (+5.4) on TextVQA and lower (-3.29) on VQAv2. Note that Flamingo's model size is 80B, which is 114 times of ours (0.7B). On VQAv2, we observe that our model performs worse in 1.5 points than the discriminative model of Florence (Yuan et al., 2021), which shares the same image encoder. The reason might be the increased difficulty of the generative model. That is, each correct answer requires at least two correct predictions (answer and [EOS]; 2.2 on average), while the discriminative model requires only one correct prediction. In (Wang et al., 2021b), the ablation study also shows the better performance by around 1 point than the discriminative counterpart. Another reason could be that the model of Florence for VQA leverages RoBerta (Liu et al., 2019) as the text encoder, which implicitly uses the text-only data to improve the performance.

## 4.3 Results on Video Captioning and Question Answering

On the video captioning task, the performance is evaluated on MSVD (Chen & Dolan, 2011) with the widely-used splits from Venugopalan et al. (2014), MSRVTT (Xu et al., 2016), YouCook2 (Zhou et al., 2018) (results in supplementary materials.) VATEX (Wang et al., 2019b), and TVC (Lei et al., 2020) (results in supplementary materials.). On VATEX, the performance is evaluated on both the public test and private test (evaluated on the server). Video QA is evaluated on MSVD-QA (Xu et al., 2017; Chen & Dolan, 2011), MSRVTT-QA (Xu et al., 2017; 2016), and TGIF-Frame (Jang et al., 2017), which are all open-ended tasks. The results are shown in Table 5 and Table 6 for captioning and QA, respectively. Although our model is not

Table 4: Results on visual question answering. (a): for VQAv2, approaches are divided according to whether the answer vocabulary is pre-defined (Closed) or not (Open) during inference. The model with closed vocabulary can be a classification model or generation model with constrained outputs, *e.g.*, Wang et al. (2022b); Li et al. (2022b). The two numbers in parenthesis are the number of parameters and the number of images (the images for pre-trained modules are not counted) in VL pretraining. (b): for TextVQA, Mia (Qiao et al., 2021)[#] is the winner entry of TextVQA Challenge 2021 with a fine-tuned T5-3B (Raffel et al., 2020) model. (c): [##]: winner entry of 2021 VizWiz Grand Challenge Workshop. ALBEF: Li et al. (2021a), BLIP: Li et al. (2022b), BLOCK+CNN+W2V: Mishra et al. (2019), CLIP-ViL: Shen et al. (2021), CoCa: Yu et al. (2022), CRN: Liu et al. (2020a), Flamingo: Alayrac et al. (2022), Florence: Yuan et al. (2021), LaAP-Net: Han et al. (2020), LaTr: Biten et al. (2022), M4C: Hu et al. (2020), M4C: Hu et al. (2020), METER: Dou et al. (2021), Mia: Qiao et al. (2021), mPlug: Li et al. (2022a), OSCAR: (Li et al., 2020), OFA: Wang et al. (2022b), UFO: Wang et al. (2021a), UNITER: (Chen et al., 2020b), UNIMO: Li et al. (2021c), SA-M4C: Kant et al. (2020), SimVLM: Wang et al. (2021b), SMA Gao et al. (2020), SMA: Gao et al. (2020), TAP: Yang et al. (2021c), VinVL: Zhang et al. (2021a), VILLA: Gan et al. (2020).

| Vocabulary | Method | test-std |
|---|---|---|
| Closed | OSCAR | 73.82 |
| | UNITER | 74.02 |
| | VILLA | 74.87 |
| | UNIMO | 75.27 |
| | ALBEF | 76.04 |
| | VinVL | 76.60 |
| | UFO | 76.76 |
| | CLIP-ViL | 76.70 |
| | METER | 77.64 |
| | BLIP | 78.32 |
| | SimVLM (-, 1.8B) | 80.34 |
| | Florence (0.9B, 14M) | 80.36 |
| | mPlug (0.6B, 14M) | 81.26 |
| | OFA (0.9B, 54M) | 82.0 |
| | CoCa (2.1B, 4.8B) | **82.3** |
| Open | Flamingo (80B, 2.3B) | **82.1** |
| | GIT (0.7B, 0.8B) | 78.81 |

(a) VQAv2

| Method | test |
|---|---|
| M4C | 40.46 |
| LaAP-Net | 41.41 |
| SA-M4C | 44.6 |
| SMA | 45.51 |
| TAP | 53.97 |
| Flamingo | 54.1 |
| Mia | **73.67** |
| GIT | 59.75 |

(b) TextVQA

| Method | test |
|---|---|
| (Liu et al., 2021)[##] | 60.6 |
| Flamingo | 65.4 |
| GIT | **67.5** |

(c) VizWiz-QA

| Method | Test ANLS |
|---|---|
| M4C | 46.2 |
| SMA | 46.6 |
| CRN | 48.3 |
| LaAP-Net | 48.5 |
| SA-M4C | 50.4 |
| TAP | 59.7 |
| LaTr | **69.6** |
| GIT | **69.6** |

(d) ST-VQA

| Method | test |
|---|---|
| BLOCK+CNN+W2V | 48.3 |
| M4C | 63.9 |
| LaAP-Net | 64.1 |
| LaTr | 67.9 |
| GIT | **68.1** |

(e) OCR-VQA

dedicated for video tasks, our model achieve new SOTA on MSRVD, MSRVTT, and VATEX for captioning and on MSVD-QA and TGIF-Frame for QA. For example on VATEX private test, our results are even better (93.8 vs 86.5) than CLIP4Caption++ (Tang et al., 2021), which relies on model ensemble and additional subtitle input. This is also better than Flamingo (Alayrac et al., 2022) (84.2) with 80B parameters.

## 4.4 Results on Image Classification

We fine-tune GIT on ImageNet-1k. Each category is mapped to a unique class name, and the prediction is correct only if it is exactly matched with the ground-truth label subject to more or fewer whitespaces[3]. As shown in Table 7, our approach can achieve descent accuracy without pre-defining the vocabulary. Compared with Florence (Yuan et al., 2021) (same image encoder), our approach is worse in about 1.2 points. The reason might be similar to the case on VQAv2. That is, the generative approach needs to predict more tokens correctly to make one correct prediction, which increases the difficulty.

**Zero-shot/Few-shot.** The result is shown in Table 9. With no knowledge of the vocabulary, the pretrained GIT cannot infer the expected vocabulary, and thus the exactly-match accuracy is only 1.93% (in the column of *equal*). However, if we relax the requirement and take it correct if the prediction contains the ground-truth, the accuracy is 40.88% (in the column of *in*), which shows the predicted caption can well identify the image content. If we have the vocabulary as a prior and limit the output tokens to be within the vocabulary, the accuracy drops to 33.48% (in the column of *voc-prior*). This may suggest the network is less natural to

---

[3]pred.replace(' ', '') == gt.replace(' ', '')

Table 5: Results on video captioning. $^E$: model ensemble; $^T$: with the subtitle as additional input. C.4Cap.: Tang et al. (2021) GRU-EVE: Aafaq et al. (2019) MGSA: Chen & Jiang (2019) MGSA: Chen & Jiang (2019) MV-GPT: Seo et al. (2022) PickNet: Chen et al. (2018) PMI-CAP: Chen et al. (2020a) SibNet: Liu et al. (2020b) OA-BTG: Zhang & Peng (2019) ORG-TRL: Zhang et al. (2020) OpenBook: Zhang et al. (2021b) POS+VCT: Hou et al. (2019) POS+CG: Wang et al. (2019a) SAAT: Zheng et al. (2020), STG-KD: Pan et al. (2020) SwinBERT: Lin et al. (2021) Support-set: Patrick et al. (2021) VaTeX: Wang et al. (2019b) VALUE: Li et al. (2021b)

| Method | B@4 | C |
|---|---|---|
| PickNet | 52.3 | 76.5 |
| GRU-EVE | 47.9 | 78.1 |
| SAAT | 46.5 | 81.0 |
| MGSA | 53.4 | 86.7 |
| POS+VCT | 52.8 | 87.8 |
| SibNet | 54.2 | 88.2 |
| POS+CG | 52.5 | 88.7 |
| OA-BTG | 56.9 | 90.6 |
| STG-KD | 52.2 | 93.0 |
| PMI-CAP | 54.6 | 95.1 |
| ORG-TRL | 54.3 | 95.2 |
| SwinBERT | 58.2 | 120.6 |
| GIT | **79.5** | **180.2** |

(a) MSVD

| Method | B@4 | C |
|---|---|---|
| SAAT | 39.9 | 51.0 |
| MGSA | 42.4 | 47.5 |
| POS+VCT | 42.3 | 49.1 |
| SibNet | 40.9 | 47.5 |
| POS+CG | 42.0 | 48.7 |
| OA-BTG | 41.4 | 46.9 |
| STG-KD | 40.5 | 47.1 |
| Support-set | 38.9 | 48.6 |
| PMI-CAP | 42.1 | 49.4 |
| ORG-TRL | 43.6 | 50.9 |
| OpenBook | 33.9 | 52.9 |
| SwinBERT | 41.9 | 53.8 |
| MV-GPT$^T$ | 48.9 | 60 |
| GIT | **53.8** | **73.9** |

(b) MSRVTT

| Method | C |
|---|---|
| VaTeX | 45.1 |
| OpenBook | 57.5 |
| VALUE$^T$ | 58.1 |
| SwinBERT | 73.0 |
| C.4Cap.$^{ET}$ | 85.7 |
| GIT | **91.5** |

(a) VATEX public test

| Method | C |
|---|---|
| X-L.+T.$^E$ | 81.4 |
| Flamingo | 84.2 |
| C.4Cap.$^{ET}$ | 86.5 |
| GIT | **93.8** |

(e) VATEX private test

Table 6: Results on video question answering. All are open-ended question answering tasks. All-in-one: Wang et al. (2022a), ClipBERT: Lei et al. (2021), CoMVT: Seo et al. (2021), Flamingo: Alayrac et al. (2022), JustAsk: Yang et al. (2021a), MERLOT: Zellers et al. (2021), MV-GPT: Seo et al. (2022), QueST: Jiang et al. (2020), HCRN: Le et al. (2021), VIOLET: Fu et al. (2021).

| Method | Accuracy |
|---|---|
| QueST | 34.6 |
| HCRN | 36.1 |
| CoMVT | 42.6 |
| JustAsk | 46.3 |
| VIOLET | 47.9 |
| All-in-one | 48.3 |
| GIT | **56.8** |

(a) MSVD-QA

| Method | Accuracy |
|---|---|
| JustAsk | 41.5 |
| MV-GPT | 41.7 |
| MERLOT | 43.1 |
| VIOLET | 43.9 |
| All-in-one | 46.8 |
| Flamingo | **47.4** |
| GIT | 43.2 |

(b) MSRVTT-QA

| Method | Accuracy |
|---|---|
| HCRN | 55.9 |
| QueST | 59.7 |
| ClipBERT | 60.3 |
| All-in-one | 66.3 |
| VIOLET | 68.9 |
| MERLOT | 69.5 |
| GIT | **72.8** |

(c) TGIF-Frame

directly predict the category name. By fine-tuning the model with only 1 shot or 5 shots per category, we observe that the accuracy is significantly improved. This demonstrates our model can be easily adapted to downstream tasks even with a few training samples. With the shot increased from 1 to 5, the gap between *voc-prior* and the other two columns (*equal* and *in*) becomes smaller. This is expected as more shots can be better to guide the network to predict in-vocabulary output.

Compared with Flamingo, our GIT achieves higher accuracy. Flamingo conducts the few-shot learning without parameter update, but each test image is combined with the support training examples as extra network inputs. Meanwhile, different test image requires different support shots based on Yang et al. (2022). These may increase the inference cost. In contrast, our model updates the parameters by a lightweight fine-tuning once, and then all these training shots are not required during inference.

## 4.5 Results on Scene Text Recognition

The task (Graves et al., 2006) aims to read scene text directly from the image. We evaluate our model in two settings. One is the GIT fine-tuned on TextCaps. The prediction is considered correct if the caption

Table 7: Results on ImageNet-1k classification task. Our approach takes the class name as the caption and predict the label in an auto-regressive way without pre-defining the vocabulary.

| Vocabulary | Method | Top-1 |
|---|---|---|
| Closed | ALIGN (Jia et al., 2021) | 88.64 |
| | Florence (Yuan et al., 2021) | 90.05 |
| | CoCa (Yu et al., 2022) | **91.0** |
| Open | GIT | 88.79 |

Table 8: Results on scene text recognition. MJ and ST indicate the MJSynth (MJ) (Jaderberg et al., 2014; 2016) and SynthText (ST) (Gupta et al., 2016) datasets used for training scene text recognition models.

| Method | FT data | Average |
|---|---|---|
| SAM (Liao et al., 2019) | MJ+ST | 87.8 |
| Ro.Scanner (Yue et al., 2020) | MJ+ST | 87.5 |
| SRN (Yu et al., 2020) | MJ+ST | 89.6 |
| ABINet (Fang et al., 2021a) | MJ+ST | 91.9 |
| S-GTR (He et al., 2022) | MJ+ST | 91.9 |
| MaskOCR (Lyu et al., 2022) | MJ+ST | **93.8** |
| GIT | TextCaps | 89.9 |
| | MJ+ST | 92.9 |

Table 9: Zero/Few-shot evaluation on ImageNet with 3 metrics. *equal*: the unrestricted prediction should be exactly matched to the ground-truth. *in*: the unrestricted prediction should contain the ground-truth label name. *voc-prior*: the vocabulary is pre-defined as a prior. For our GIT, a trie structure is constructed motivated from Wang et al. (2022b) to limit the candidate tokens during each token prediction, such that the predicted result is guaranteed to be within the vocabulary.

| Accuracy type | Zero-shot | | | 1-shot per class | | | 5-shot per class | | |
|---|---|---|---|---|---|---|---|---|---|
| | equal | in | voc-prior | equal | in | voc-prior | equal | in | voc-prior |
| Flamingo | - | - | - | - | - | 71.7 | - | - | 77.3 |
| GIT | 1.93 | 40.88 | 33.48 | 64.54 | 66.76 | 72.45 | 79.79 | 80.15 | 80.95 |

contains the ground-truth scene text word. The other is to fine-tune the model on two large scene text datasets: MJSynth (MJ) (Jaderberg et al., 2014; 2016) and SynthText (ST) (Gupta et al., 2016), where the ground-truth scene text is used as the *caption*. The prediction is correct if the output is the exact match to the ground-truth. Following the established setup, we evaluate on six standard benchmarks, including ICDAR 2013 (IC13) (Karatzas et al., 2013), ICDAR 2015 (IC15) (Karatzas et al., 2015), IIIT 5K-Words (IIIT) (Mishra et al., 2012), Street View Text (SVT) (Wang et al., 2011), Street View Text-Perspective (SVTP) (Phan et al., 2013), and CUTE80 (CUTE) (Risnumawan et al., 2014). The average accuracy is reported in Table 8. The accuracy on individual test sets is in supplementary materials. Our TextCaps-fine-tuned captioning model achieves an 89.9 accuracy, which demonstrates the strong scene text comprehension capability of our captioning model. After fine-tuning the model on the standard MJ+ST datasets, GIT achieves 92.9 that surpasses the prior arts (Fang et al., 2021a; He et al., 2022) of 91.9.

### 4.6 Analysis

**Model and data scaling.** To study the trending with data scales, we construct two smaller pre-training datasets: one is the combination of COCO, SBU, CC3M and VG, leading to 4M images or 10M image-text pairs; the other is to further combine CC12M, leading to about 14M images or 20M image-text pairs. When pre-training on small-scale datasets, we use 30 epochs rather than 2 epochs as on the 0.8B data. For the network structure, we name our model as *Huge* and replace the image encoder with ViT-B/16 and ViT-L/14 from CLIP Radford et al. (2021) as *Base* and *Large*, respectively. Fig. 4 shows the results on COCO, TextCaps, and VizWiz-QA. On COCO, the base model benefits from 4M to 14M, but the performance drops with 0.8B data. The 14M data are more similar to COCO than the majority of the noisy 0.8B data. Meanwhile, the Base model with limited capacity may not be able to benefit effectively from large-scale data. Similar observations are also reported in Kolesnikov et al. (2020) for ImageNet-1k classification. On TextCaps and VizWiz-QA, all model variants benefit significantly from more pre-training data. Also, a larger backbone improves more especially with 0.8B data.

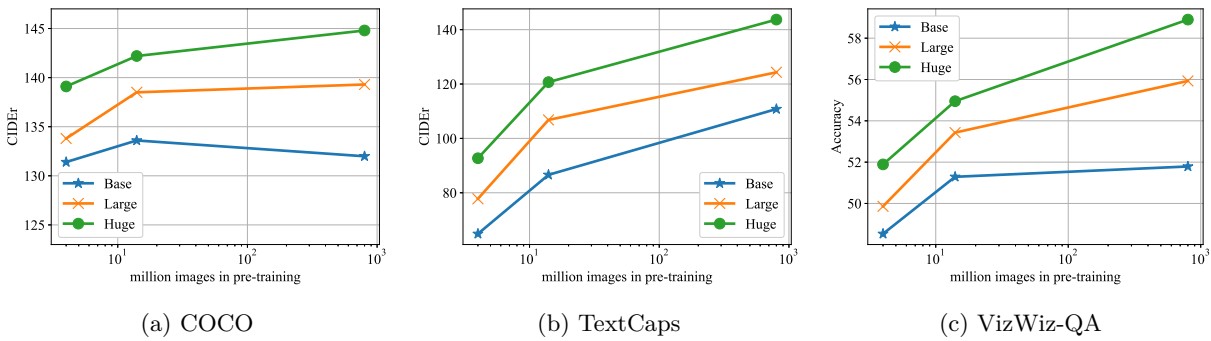

(a) COCO                    (b) TextCaps                    (c) VizWiz-QA

Figure 4: Performance with different pre-training data scales and different model sizes.

Table 10: Ablation study of larger text decoders. The models are pre-trained on a subset of 0.4B image-text pairs. No beam search and no SCST are performed.

| Layers | COCO | | | | nocaps | |
|---|---|---|---|---|---|---|
| | B@4 | M | C | S | C | S |
| 6 | 38.9 | 30.7 | 136.4 | 24.6 | 119.3 | 15.9 |
| 12 | 38.9 | 30.6 | 136.0 | 24.2 | 118.1 | 15.5 |
| 24 | 39.1 | 30.2 | 134.6 | 23.8 | 115.4 | 15.1 |

Here, we scale the image encoder. Empirically, we find it is difficult to effectively scale up the text decoder. Preliminary results are shown in Table 10, which shows a larger decoder shows no improvement. The reason might be that it is difficult to effectively train with limited amount of text by LM. Another plausible reason is that the image encoder is responsible for object recognition, and the decoder is responsible for organizing the object terms in a natural language way. The latter task might be easy since most of the descriptions follow similar patterns, e.g. object + verb + subject, and thus a small decoder is enough during end-to-end training. Larger decoders increase the learning difficulty, which might degrade the performance.

Flamingo (Alayrac et al., 2022) shows a larger decoder improves the performance. However, their decoder is pre-trained and frozen during the VL pre-training, which avoids the problem of how to effectively train the decoder. In LEMON (Hu et al., 2021a), the transformer can be scaled up to 32 layers. The reason could be that LEMON uses MLM, instead of LM, which might be more difficult to train.

**Scene text in pre-training data.** To understand the capability of scene text comprehension, we examine the pre-training dataset and study how many image-text pairs contain the scene text. We first run the Microsoft Azure OCR API[4] against all images in CC12M and 500K images in the web crawled images. The OCR result is compared with the associated text. It is considered *matched* only if the text contains an OCR result that is longer than 5 characters. It is estimated that 15% of CC12M and 31% of the downloaded images contain scene text descriptions. As the training task is to predict the texts, the network gradually learns to read the scene text.

## 5 Conclusion

In the paper, we design and train a simple generative model, named GIT, to map the input image to the associated text description on large-scale image-text pairs. On image/video captioning and question answering tasks, our model achieves new state-of-the-art performance across numerous benchmarks and surpasses the human performance on TextCaps for the first time. For the image classification, we apply the generation task to predict the label name directly. The strategy is different from the existing work with a pre-defined and fixed vocabulary, and is beneficial especially when new category data are added.

---

[4]https://docs.microsoft.com/en-us/azure/cognitive-services/computer-vision/concept-recognizing-text

**Limitations.** We focus on the pretraining-and-finetuning strategy to improve the absolute performance. Empirically, we find it is unclear on how to control the generated caption and how to perform in-context learning without parameter update, which we leave as future work.

**Societal impact.** Compared with the existing work, our model clearly improves the performance and be more appropriate to help visually-impaired people. The model is pre-trained on large-scale data, and the data are not guaranteed to contain no toxic language, which may poison the output. Although we observe few such instances qualitatively, special care should be taken to deploy the model in practice and more research exploration is required to control the output.

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
