# OpenReview forum: "GIT: A Generative Image-to-text Transformer for Vision and Language"
_TMLR — Accepted by TMLR_

### Review · Reviewer_6mzR · 2022-09-11

**Summary Of Contributions:**

The paper presents a new architecture GIT which pose a diverse range of vision-and-language tasks as a generative language modeling task. The authors show posing all tasks as a single language modeling task has many desirable properties – simplification of training paradigm (pre-training and fine-tuning on downstream task is consistent), data can be scaled for pre-training and there is no need for expert modules like OCR, object detection. Using GIT, authors show state-of-the-art performance on many vision and vision-and-language tasks including tasks that require reading text in image.

**Requested Changes:**

I am tending to recommend the paper for acceptance, but it'd be nice to include these results
- Few-shot results similar to Flamingo, CoCa on the various benchmarks. While these won't come very close to the sota results, it'd still be good to see what few-shot learning performance looks like.


**Strengths And Weaknesses:**

## Strengths:
- Large-scale pretrained vision-and-language mode have been very successful and has seen a lot of innovation in the last couple of years. But existing methods require careful pre-training, multiple training objectives and discriminative tasks (VQA, etc) and image-captioning tasks are handled differently. This paper presents a minimalist approach – treat all tasks as generative language modeling tasks and show impressive performance on a range of downstream tasks.
- Overall results are great, especially the results on nocaps and TextCaps! I enjoyed seeing the qualitative results present in the supplementary.
- I was pleasantly surprised by how well the model works on the TextCaps dataset both qualitatively and quantitatively. Without using the output of a OCR module as input, the model was able to produce captions that correctly recognized text present in the image, and use it in the output caption.

## Weaknesses:
1. As mentioned in the manuscript, one of the limitations of the approach is the requirement of fine-tuning on a range of tasks. The paper doesn't show few-shot results on any of the tasks.
  1.1 Specifically, on tasks like COCO, nocaps, being trained on 0.8 billion image-text pairs, it would have been nice to see zero-shot/few-shot performance at least on the captioning tasks.
  1.2 Similarly, tasks like image classification can also be evaluated in a zero-shot/few-shot manner, right? It would have been great to see these results in the paper
  1.3 Flamingo (Table 6) (https://arxiv.org/pdf/2204.14198.pdf) show few-shot results on a range of benchmarks, and it would be good to discuss the model's ability for few-shot learning.

2. Discussion of failure modes: While the authors discuss reasons for why the performance is a few points lower than other methods (e.g., VQA performance is lower because of posing the task as a generative task vs discriminative), I would have liked to see some discussion of what authors think are fundamental limitations of the paper. For instance, if this model is not capable of few-shot learning, why is that?

3. I'd also like to point out that the author's evaluation on nocaps is not consistent with the guidelines in the benchmark. According to the guidelines in the original paper, large-scale training on additional image-caption pairs that don't come from COCO should not be used. The authors of the nocaps benchmark, specifically created a server for this usecase (nocaps-XD) which should be used when using extra image-caption paired data.

3. A very minor comment on the supplementary – the text, tables and figures are put together in a way that is hard to read. For instance, 2 lines of the previous section show up after the big table 7 on page 19. I recommend reformatting it such that text for each section appear in one place.

---

> ### Author Response · Authors · 2022-09-27
> **Response to Reviewer 6mzR**
>
> Thanks for the insightful reviews and the praise for our simple yet effective approach to a range of VL tasks. We are also glad the reviewer enjoys seeing the qualitative results present in the supplementary, which is as important as the SOTA numbers to understand the behavior of how the model works. Here, we address other concerns as follows.
>
> -------------------------------------
>
> Q1: Zero-shot/few-shot performance
>
> A1: Thanks for the suggestions. We added the zero-shot/few-shot performance for image captioning and imagenet classifications in the manuscript. Specifically, on the image captioning, we evaluate the performance on Flickr with the Karpathy split. We do not evaluate it on COCO or nocaps because the COCO dataset are already in the pretraining dataset. The result is shown as follows, in which we can observe 1) with few-shots of 16 or 32 shots, our approach is better than Flamingo, 2) our approach benefits from more shots from 16 to 32, while it is not necessary to be the case for Flamingo, which is the in-context learning without parameter updates. Last but not the least, with full fine-tuning, we can also achieve new SoTA performance on this dataset.
>
> |                   | 0    | 16   |  32   | 290 (1%) | full |
> |-------------------|------|------|-------|----------|------|
> | Zhou et al (2020) | -    | -    | -     | -        | 68.5 |
> | Flamingo          | 67.2 | 78.9 | 75.4  |          |      |
> | GIT               | 49.6 | 78.0 | 80.5  | 86.6     | 98.5 |
>
>
> The following shows the results of the zero/few-shot study on ImageNet. As our model is generative, which can predict any output, we evaluate the performance in three metrics: 1) the predicted caption should be exactly matched to the ground-truth label name, named as equal; 2) the predicted caption should contain the ground-truth label name, named as in; 3) with the vocabulary as a prior, we apply a trie structure to constrain the candidate tokens in each prediction step, such that the predicted caption is always within the vocabulary. As only the output space is limited from the whole token vocabulary to a subset, the inference cost is almost not changed, and is also constant to the category size. For Flamingo, each category label is fed into the network to compute the score and is thus linear to the vocabulary size. From the results, we can see with the vocabulary as a prior, our generative model can be better than Flamingo with 1 shot or 5 shot per class. More discussions are added in the paper (Sec. 4.5, table 9; supplementary material:table 13)
>
>
> |               |       | zero-shot |              |       | 1 shot per class |              |       | 5 shot |           |
> |---------------|-------|-----------|--------------|-------|------------------|--------------|-------|--------|-----------|
> | Accuracy type | equal | in        | voc-prior    | equal | in               | voc-prior    | equal | in     | voc-prior |
> | Flamingo      |       |           |              |       |                  | 71.7         |       |        | 77.3      |
> | GIT           | 1.93  | 40.88     | 33.48        | 64.54 | 66.76            | 72.45        | 79.79 | 80.15  | 80.93     |
>
>
> Q2: Failure mode
>
> A2:  The design is specifically for the pretraining-and-finetuning scenarios. It is unclear how to apply the in-context learning such that the network can adapt to the new task without parameter update. In the meanwhile, we also find it is not clear how to control the output style. For example, we cannot control it should always output the scene text or not to output the scene text even if it exists in the image. For the few-shot, recently we find out the capability is also strong as discussed in A1 for the first question.
>
> Q3: nocaps protocol
>
> A3: Thanks for pointing out this. We added footnote 2 in the paper to make it clear.
>
> Q4: text format for the supplementary materials
>
> A4: Thanks for the suggestions, and we have reformated the text.

---

### Review · Reviewer_hDXf · 2022-09-13

**Summary Of Contributions:**

**Contributions**

The paper proposes a unified architecture GIT for vision and language tasks. The GIT model is able to perform well across a variety of vision and language tasks like video/image captioning, video/image question answering and scene text recognition without relying on external feature extractors like object detectors and OCR. The model performs significantly well on tasks which involve reasoning over scene text in images. The approach used is simple and scales well with more pre-training data and the model size.

**Broader Impact Concerns:**

 The model is pre-trained on large-scale data, and the data used for training might contain toxic content and biases therefore the model might generate predictions which contain biases or toxic language.

**Requested Changes:**

Highlighted in the weaknesses section above.

**Decision**

I am recommending accept as the authors have provided requested changes in feedback.

**Strengths And Weaknesses:**

**Strengths**

1. The authors propose an end to end model capable of performing well across a variety of image and captioning tasks.

2. The architecture proposed is simple and the model is trained using a single pre-training task instead of relying on more supervision and multiple pre-training tasks.

3. The experimental section of the paper is clear and the authors have specified all the details regarding the various experiment and analysis conducted.

4. In general, the paper is well written and is easy to follow.

**Weaknesses**

1. Although the results in section 4.5 showcase that the model is able to achieve good performance on standard OCR datasets, it is still not clear how the model learns to do OCR well. Is the model able to do OCR well by just adding more images which contains scene text in it, or is it the architecture of the model that contributes to good OCR performance? Also, the paper does not contain experiment on a recent TextOCR dataset.

3. To compare GIT to recent large scale vision and language models, instead of just comparing performance the authors should also compare the amount of training data (pre-training and fine-tuning) and the number of parameters used by the baselines and GIT. Including this analysis might strengthen the paper further.

---

> ### Author Response · Authors · 2022-09-27
> **Response to Reviewer hDXf**
>
> Thanks for the valuable comments and the praise for our approach's simplicity and clear paper presentation. Here, we address other concerns as follows.
>
> ------------------------------------------
> Q1: How the model learns OCR capability.
>
> A1: The model structure only consists of one image encoder and one text decoder. There are no OCR-specific modules. The OCR capability is contributed most by the data, we believe. As discussed in the second part of Sec. 4.6, we observe 15\% to 31\% of the associated texts, which contains the scene text. As the model structure is to learn the mapping from the image to the associated text, we believe the network we designed learns such OCR capability from the data.
>
> Q1.1: Is the model able to do OCR well by just adding more images which contains scene text in it, or is it the architecture of the model that contributes to good OCR performance.
>
> A1.1:   Our model structure is to learn the mapping from the image to the associated text. The data contains the scene text in the text. Thus, we believe both the model structure and the data are important to the OCR capability, and the data should be the one which contributes the most.
>
> Q1.2: The paper does not contain experiment on a recent TextOCR dataset
>
> A1.2: From the official website of TextOCR (https://textvqa.org/textocr/), we find out there are no test sets publicly available. Although the validation set is released in the website, the reference paper of TextOCR only reports the results on the test set rather than on the validation set. We fine-tuned our pretrained GIT model on TextOCR with 10 epochs and 2e-5 as the learning rate. The accuracy on the validation set is 81.27. In the paper of TextOCR, the reported best accuracy on the test set is 69.49. If we assume the validation set and the test set follow similar distribution, we believe our approach without any dedicated OCR modules can also achieve competitive or better performance on this dataset.
>
> Q2: Training data (pre-training and fine-tuning) and model parameters comparison.
>
> A2: In terms of the training data, all methods share the same amount of fine-tuning data. The training data and the model parameters are also reported in the Table 2 of the supplementary materials. Roughly, the performance improves with more training data and more model parameters. For the base or the large-sized model, our model's performance is competitive to the existing work. For example on the base-sized model, our model achieves 131.4 CIDer score with 4M pretraining images, while VinVL (a representative approach of VL model based on object detection) achieves 129.3 with 6M pretraining images.

---

> > ### Comment · Action_Editors · 2022-09-28
> > **Adding TextOCR results to the paper?**
> >
> > Dear authors, are you considering adding the TextOCR results to the paper?

---

> > > ### Author Response · Authors · 2022-09-29
> > > **TextOCR results added**
> > >
> > > The results of TextOCR are added in Appendix F in the updated submission.

---

### Review · Reviewer_q1NN · 2022-09-15

**Summary Of Contributions:**

In this paper, the authors proposed a simple generative image-to-text framework with a unified encoder-decoder design for various downstream tasks. New state-of-the-art is achieved on various downstream tasks. This framework is also shown to be effective on directly generating image class names for image classification.

**Broader Impact Concerns:**

The open-vocabulary generation property makes this proposed foundation model potentially easier to have negative impact. Therefore, it will be interesting and somewhat necessary to further evaluate the bias learned in this model. For example, a further test could be evaluating the model for image captioning and see if it has severe issue of racial bias [r5].

[r5] Understanding and Evaluating Racial Biases in Image Captioning, ICCV 2021

**Requested Changes:**

Please check the cons and broader impact concerns for details of required new items.

Post-discussion:
Thanks for the new results and discussions! But the reviewer noticed that many of the discussions are not added to either the main paper or the supplementary in the revision.

The reviewer tends to recommend acceptance if all the discussions are added. If the reviewer missed anything, it would be very helpful if the authors could provide a change-list for the revisions.

**Strengths And Weaknesses:**

Pros:
1. The proposed framework is simple yet effective.

2. Very strong empirical results.

3. The paper is presented overall clearly.

Cons:

1. It is hard to understand the true source of the improvement in comparisons with existing methods. Due to the difference in pretraining data & its size, model size, number of pretrained encoders/decoders used and fine-tuning data & its size, from the main SOTA comparison Tables, it is impossible to understand the possible reasons leading to the improvements. Therefore, it is important to at least add details from these aspects for the most competitive compared methods to clarify the main source of improvements.

2. Lack further insightful analysis of the behavior of the model.

- a. The text decoder is randomly initialized but there is no analysis on the behavior of the model when the text decoder is initialized from some pretrained LMs. This is actually counter-intuitive and very different from existing work where leveraging pretrained text decoder can even avoid the vision-text pretraining process [r1,r2,r3,r4]. The authors should discuss this different behavior of the proposed model and show more insights on the possible reasons.

- b. The different trend in terms of the size between image encoder and text decoder. In Flamingo, text decoder is much larger than the image encoder but in this work the image encoder is usually larger than the text decoder. Is this correlated to the fact that currently the text decoder is randomly initialized? What is the principle on the deciding the size of each component? What is the performance when using a relatively much larger text decoder?

- c. The reviewer understands that the authors have genuinely listed the zero/few-shot ability as a limitation. But it is still very interesting to see a comprehensive performance comparison to understand the gain of fine-tuning. This could be also helpful to understand the benefit from different fine-tuning datasets.

3. Minor: the details of how videos are sampled/preprocessed is missing.

[r1] VX2TEXT: End-to-End Learning of Video-Based Text Generation From Multimodal Inputs, CVPR 2021

[r2] Socratic Models: Composing Zero-Shot Multimodal Reasoning with Language

[r3] Visual Clues: Bridging Vision and Language Foundations for Image Paragraph Captioning

[r4] Language Models with Image Descriptors are Strong Few-Shot Video-Language Learners

---

> ### Author Response · Authors · 2022-09-27
> **Response to Reviewer q1NN (1/2)**
>
> Thanks for the valuable comments and the praise for the strong performance of our approach with a simple generative task. Here, we address other concerns as follows.
>
> -------------------------------
>
> Q1: source of the improvement in comparisons with existing methods
>
> A1: The key source of the improvement is large data scales and large model size. In Sec 4.6 and Supplementary G.1, we show the improvement with the data scaling and model scaling on 9 benchmarks in total, which reveals scaling can significantly improve the performance. In Table 2 of the supplementary materials, we added information of the pretraining dataset scales and the parameters to compare other approaches. Our base-sized model with 4M pretraining images achieves competitive performance (131.4 CIDEr) compared with, e.g. VinVL (a representative approach of VL model based on object detection, 129.3) with 6M pretraining images.
>
> Q2.a: initialization of the text decoder
>
> A2.a: As discussed in Sec 3.1, we randomly initialize the text decoder based on the ablation observation of MiniVLM. We also added the ablation study to initialize the transformer in the updated supplementary (G.3). Specifically, the following shows the result, and we observe no improvement. The reason could be that the initialized weight cannot understand the image input, which is crucial for the VL. In Flamingo, the pretrained transformer is fixed and thus the initialization is important. In VX2TEXT (r1), all modalities are converted into the textual names and embedded into the same language space. The input to the transformer is essentially the text, rather than the image feature as what we do. Even without any training, the input is meaningful to the pretrained transformer and thus it is reasonable to use the pretrained decoder rather than to initialize randomly. In (r2), (r3) and (r4), the models are not trained, and thus the weight is important to be initialized properly to equip the model respective capability.
>
>
> | Initialization  | COCO  | TextCaps  | VizWiz-Captions |
> |-----------------|-------|-----------|-----------------|
> | Random          | 127.5 | 61.7      | 68.0            |
> | Initialized     | 126.9 | 59.7      | 67.0            |
>
> Q2.b: Flamingo uses larger text decoder. Is this correlated to the fact that currently the text decoder is randomly initialized? What is the principle on the deciding the size of each component? What is the performance when using a relatively much larger text decoder?
>
> A2.b: We agree on the point that as we randomly initialize the decoder, it is more difficult to train the component effectively, while Flamingo freezes the decoder, which circumvents the problem of how to train the decoder. We also present the experiment results on enlarging the decoder in Table 8 of the main paper, and empirically we do not observe improvement with larger decoders. The golden principle on the module size of the decoder is to have the experiment results, we think. Based on the current results and the existing literature, it seems like we can conclude that if the decoder is trained together with the whole network, it should be small; if it is fixed, it should be well initialized and should be as large as possible.

---

> ### Author Response · Authors · 2022-09-27
> **Response to Reviewer q1NN (2/2)**
>
> Q2.c: zero/few-shot capability
>
> A2.c: Thanks for the suggestions. We added the zero-shot/few-shot performance for image captioning and imagenet classifications in the manuscript. Specifically, on the image captioning, we evaluate the performance on Flickr with the Karpathy split. We do not evaluate it on COCO or nocaps because the COCO dataset are already in the pretraining dataset. The result is shown as follows, in which we can observe 1) with few-shots of 16 or 32 shots, our approach is better than Flamingo, 2) our approach benefits from more shots from 16 to 32, while it is not necessary to be the case for Flamingo, which is the in-context learning without parameter updates. Last but not the least, with full fine-tuning, we can also achieve new SoTA performance on this dataset.
>
> |                   | 0    | 16   |   32 | 290 (1%) | full |
> |-------------------|------|------|-------|----------|------|
> | Zhou et al (2020) | -    | -    | -     | -        | 68.5 |
> | Flamingo          | 67.2 | 78.9 | 75.4  |          |      |
> | GIT               | 49.6 | 78.0 | 80.5  | 86.6     | 98.5 |
>
>
> The following shows the results of the zero/few-shot study on ImageNet. As our model is generative, which can predict any output, we evaluate the performance in three metrics: 1) the predicted caption should be exactly matched to the ground-truth label name, named as equal; 2) the predicted caption should contain the ground-truth label name, named as in; 3) with the vocabulary as a prior, we apply a trie structure to constrain the candidate tokens in each prediction step, such that the predicted caption is always within the vocabulary. As only the output space is limited from the whole token vocabulary to a subset, the inference cost is almost not changed, and is also constant to the category size. For Flamingo, each category label is fed into the network to compute the score and is thus linear to the vocabulary size. From the results, we can see with the vocabulary as a prior, our generative model can be better than Flamingo with 1 shot or 5 shot per class. More discussions are added in the paper (Sec. 4.5, table 9; supplementary material:table 13)
>
>
> |               |       | zero-shot |              |       | 1 shot per class |              |       | 5 shot |           |
> |---------------|-------|-----------|--------------|-------|------------------|--------------|-------|--------|-----------|
> | Accuracy type | equal | in        | voc-prior    | equal | in               | voc-prior    | equal | in     | voc-prior |
> | Flamingo      |       |           |              |       |                  | 71.7         |       |        | 77.3      |
> | GIT           | 1.93  | 40.88     | 33.48        | 64.54 | 66.76            | 72.45        | 79.79 | 80.15  | 80.93     |
>
>
>
> Q3: The details of how videos are sampled/preprocessed is missing.
>
> A3: Thanks. We added the details in Supplementary D. That is, "during training, we randomly sample 6 frames with equal interval, and apply the same random crop on these frames. During inference, we uniformly sample 6 frames with center crop." As also with the image domain, we only apply the random crop without any other augmentation, e.g. random flip, color distortion.
>
> Q4: bias study
>
> A4: Thanks for pointing out this. We added the bias study in G.6 of the Appendix based on the reference of (r5). Specifically, (r5) provides the gender type (male or female) and the skin type (light or dark) for the COCO 2014 test images containing people. As we use the Kapathy split, we first collect the overlapped images between the Kapathy test and the  images with well-defined gender and skin annotations. Then, we evaluate the performance on the subset images of each category. To measure the bias, we calculate the normalized performance difference (NPD). For example of the gender, we first obtain the metric (e.g. CIDEr) on the images annotated with male (C1) and on the images with female (C2). Then, NPD is |C1 − C2|/(C1 + C2). With no bias, C1 should equal C2 and NPD is 0. If the model performs well on one group but totally fails on the other group (metric is 0), NPD is 1. As shown in the following table, we can see that the bias only ranges between 0.7% and 5.3% across all metrics.
>
> |        | BLEU@4 | METEOR | CIDEr | SPICE |
> |--------|--------|--------|-------|-------|
> | Gender | 0.7%   | 0.9%   | 2.0%  | 2.1%  |
> | Skin   | 4.2%   | 2.3%   | 5.3%  | 2.2%  |

---

> ### Author Response · Authors · 2022-10-13
> **Change list**
>
> Thanks for the suggestion. Here is the compiled change list. All the changes have already been added to the main paper or the supplementary materials. Please also let us know if anything is missed.
>
> - Added the experiments of zero/few/full shots on Flickr captioning task in Sec. 4.2/Table 3 of the main paper and table 7 of the supplementary materials.
> - Added the extra information of using external datasets for nocaps benchmark to reduce confusion in footnote 2.
> - Added the zero/few-shots results on ImageNet classification task in Sec 4.4/Table 9 of the main paper and table 13 of the supplementary materials.
> - Added the details on how the video frames are sampled in Sec D of the supplementary materials.
> - Added the results on TextOCR in Sec. F of the supplementary materials.
> - Added the ablation study of different initialization schemes for the text decoder transformer in Sec G.3/Table 16 of the supplementary materials.
> - Added the ablation study of the different initialization schemes for the image encoder in Sec. G.4/Table 17 of the supplementary materials.
> - Added the bias study over gender and skin in Sec G.6/Table 19 of the supplementary materials.

---

> > ### Comment · Reviewer_q1NN · 2022-10-13
> > **Missing discussions**
> >
> > Thanks for the fast response of the change list!
> > However, according to the change list, at least two interesting discussions that could possibly enlighten future research are not included: one is the comparison with existing work leveraging pretrained text transformers; the other one is the relative size of the text decoder compared to the image encoder.
> > These two questions are important design questions for future work and it would be of great help for future researchers if the authors directly identify and discuss these two questions in the paper.
> >
> > The reviewer also noticed another problem for the experiment for initializing model with pretrained text transformer weights during this discussion: currently the experiment of using initialized weights is only for BERT-base. However, as the author mentioned in the paper that using larger text transformer makes it hard to train from random initialization, it is necessary to check the performance for using pretrained larger text models to complete this discussion and make the paper self-contained.

---

> > > ### Author Response · Authors · 2022-10-20
> > > **Response to the missing two discussions and update on the new revision**
> > >
> > > For the first discussion on using the pretrained text transformers, the ablation study and the discussion is in G.3 and Table 16 of the supplementary materials. The key conclusion is that in our model design, the pretrained weight cannot give higher performance, while the related work may benefit as they freeze the transformer or use the text instead of image signals as the transformer input.
> > >
> > > For the second discussion on the size of the decoder, we discuss this issue in Sec 4.6 and Table 10 of the main paper. In G.3 and Table 16 of the supplementary materials, there are also some discussions on larger text transformer sizes. The key conclusion is that we do not observe better performance, which may be due to the difficulty of model training with larger sizes.
> > >
> > > The experiment with a larger pretrained text transformer is added in G.3 and Table 16 of the supplementary materials. Thanks for this suggestion! The newly added experiment result also shows consistent observations.
> > >
> > > Thanks again for all the great comments, and please let us know of any other concerns.

---

### Review · Reviewer_a1U9 · 2022-09-16

**Summary Of Contributions:**

The paper proposes to use a simple encoder-decoder design for pre-training a vision-and-language Transformer. The model is then fine-tuned to a number of tasks, mainly for vision-and-language understanding but also for vision-only tasks, and shows good results. It shows that the simple encoder-decoder design is powerful enough to achieve state-of-the-art.

**Broader Impact Concerns:**

Not aware of.

**Requested Changes:**

* In light of the first weakness, I am curious about the role of the pre-trained visual encoder and think more experiments can be devoted there for ablations. E.g. what if randomly initialize the visual encoder; what if initialized from supervised/self-supervised learning purely on images.


**Strengths And Weaknesses:**

+ The experiments are very extensive. With the majority of the tasks focused on vision-and-language understanding, the paper evaluates on all the vision-and-language task I am aware of, and achieves impressive results there.
+ I like the architecture simplicity, and overall direction that tries to scale the pre-training data and model size on simple models to make progress

- Against the simplicity is that approach initializes the visual encoder from image-text contrastive learning (CLIP-like). The additional stage is adding complications and is not friendly for reproducing results or better scientific understanding.
- Overall the paper lacks good ablations. This may due to the fact that a lot of the space is devoted to state-of-the-art, system-level comparisons. However, I believe analytical experiments are equally important and should be taking up more space.
- Relative to the above point is the "novelty". I understand novelty is subjective and for simple approaches it is a bit demanding. However, since there are already so many works showing vision-and-language works existing and as a reader I am not so surprised to see size matters and yet another better number is achieved with the next-scale model. Therefore, I believe insight or rigorous analysis is the key here to at least remedy the concern to an extent.

---

> ### Author Response · Authors · 2022-09-27
> **Response to Reviewer a1U9**
>
> Thanks for the valuable comments and the praise for extensive experiment results and the simplicity of our approach. Here, we address other concerns as follows.
>
> ----------------------------------------
>
> Q1: Initialization of the image encoder.
>
> A1: Thanks for pointing out this ablation study. We added it in the updated supplementary as G.4. Specifically, the following table shows the comparison with other initialization methods. The setting follows GIT-B. The image encoder is the base-sized version of ViT, which is initialized 1) from the CLIP model, 2) from the supervised pretraining (classification task on ImageNet), 3) from the self-supervised pretraining (MAE on ImageNet), or 4) randomly initialized. From the results, we can clearly observe the higher performance with the CLIP pretrained weights. Compared with the supervised/self-supervised, we note that the pretraining datasets for the image encoder are different here due to the availability of these weights. Although it is unclear whether the pretraining dataset is more important than the task or vice versa, we choose the contrastive pretraining as the pretraining dataset is also easy to scale up. For randomly initialization, we observe significant lower performance. The reason could be the small scale of the pretraining set (10M image-text pairs in the set-up of GIT-B ). A larger dataset may reduce the gap, but it may require longer training iterations. We leave how to effectively train the model from scratch as future work.
>
> |                 | COCO  | TextCaps | VizWiz-Captions |
> |-----------------|-------|----------|-----------------|
> | CLIP            | 131.4 | 64.9     | 61.2            |
> | Supervised      | 122.1 | 47.0     | 58.3            |
> | self-supervised | 123.4 | 44.9     | 51.6            |
> | Random          | 89.0  | 36.5     | 38.1            |
>
> Q2: ablation studies
>
> A2: Overall, we have the following ablation studies in the updated manuscript. 1) Sec 4.6 and G.1: scaling behavior with different model sizes and different pretraining scales on 9 datasets in total. 2) Sec 4.6: scaling up the text decoder does not show improve performance in our settings. 3) G.2: compare the pure self-attention decoder and the cross-attention-based decoder. 4) G.3: different initialization methods on the text decoder. 5) G.4: different initialization methods on the image encoder. 6) G.5: Intermediate fine-tuning on VQA which leads to better performance. 7) G.6: bias study over gender and skin of our captioning model. 8)Sec. 4.6 and G.7: scene text data analysis in the pretraining dataset.

---

### Decision · Action_Editors · 2022-11-18

**Recommendation:** Accept with minor revision

**Comment:**

The paper overall successfully supported the claims experimentally with throughout strong and extensive experimental evaluation.
Overall, all reviewer recommend accept, but with some hesitation.

Specifically, the reviewers and action editor, like to ask the author to improve the following points in a minor revision
* integrate missing discussion of the author/reviewer discussions in paper.
* add any additional insights the authors might have to provide more understanding the authors have gained in the meantime to provide more details for how the model is working and how the improvements are achieved
* thorough proof read of the paper

**Audience:**

The paper is of high interest to the multimodal learning (vision & language) community and also of interest, more generally, to the large scale deep learning community.

**Claims And Evidence:**

The submission makes the following claims which are well supported by the paper's presentation and experiments:
* A simple image encoder-decoder architecture can generalize to many task, with a generative decoder architecture
* This architecture when pre-trained with 0.8 billion image-text pairs, achieve or improve over state-of-the-art, for some tasks / datasets by large margin. This includes diverse tasks in image captioning, VQA, Video caption, video captioning, video QA, Scene Text Recognition.

---

> ### Author Response · Authors · 2022-11-28
> **Upload of Camera Ready Revision**
>
> We sincerely thank all reviewers and AE for the insightful comments! The camera-ready revision has been uploaded with all the comments addressed.